# Sustaining the Elimination of Measles, Rubella and Congenital Rubella Syndrome in the Americas, 2019–2023: From Challenges to Opportunities

**DOI:** 10.3390/vaccines12060690

**Published:** 2024-06-20

**Authors:** Gloria Rey-Benito, Desirée Pastor, Alvaro Whittembury, Regina Durón, Carmelita Pacis-Tirso, Pamela Bravo-Alcántara, Claudia Ortiz, Jon Andrus

**Affiliations:** 1Comprehensive Immunization Program, Pan American Health Organization, 525 23rd Street, NW, Washington, DC 20037, USA; reyglori@paho.org (G.R.-B.); whittemburya@paho.org (A.W.); duronreg@paho.org (R.D.); paciscar@paho.org (C.P.-T.); bravopam@paho.org (P.B.-A.); ortizcla@paho.org (C.O.); 2Department of Global Health, Milken Institute School of Public Health, George Washington University, Washington, DC 20037, USA; andrusjonkim@yahoo.com

**Keywords:** measles, rubella, vaccination coverage, epidemiological factors, laboratories, Pan American Health Organization, Americas

## Abstract

This report reviews national data from all Member States on measles, rubella, and congenital rubella syndrome (CRS) elimination in the Region of the Americas during 2019–2023. It includes an analysis of compliance with vaccination coverage, surveillance indicators, and measles outbreaks, as well as an analysis of the response capacity of the laboratory network and a country case study that meets all indicators. The sources of information were the integrated epidemiological surveillance system for measles and rubella of the Pan American Health Organization (PAHO)/World Health Organization (WHO) and the Joint Reporting Form (eJRF), among others. From 2020 to 2022, regional coverage with first (MMR-1) and second doses (MMR-2) decreased to rates below 90%. The regional suspected case notification rate was maintained above the minimum expected 2.0 suspect cases per 100,000 population, except in 2021. During 2019 to 2023, 18 countries experienced outbreaks, with two of the outbreaks resulting in re-established endemic transmission. In conclusion, two countries in the Americas have not maintained measles elimination, but by the end of 2023 no country showed endemic measles transmission. One of the countries that lost its certification of elimination in 2018 managed to be reverified in 2023; the other is pending reverification. All countries maintained rubella elimination. Despite these challenges, the sustainability of the elimination of these diseases remains a health priority in the Region.

## 1. Introduction

In 1994, the countries of the Region of the Americas, led by the Pan American Health Organization (PAHO), set the goal of interrupting endemic measles virus transmission by the year 2000 [1]. Between 1994 and 2014, these countries implemented two classic elimination strategies based on achieving high vaccination coverage (≥95%) and maintaining adequate epidemiological surveillance, including laboratory surveillance [2]. To date, the Region of the Americas has been the only region of the World Health Organization (WHO) to have achieved the elimination of endemic measles and rubella viruses, a process that was verified by an International Committee of Experts; it was also the first region to create an Action Plan to Verify the Elimination of Measles, Rubella and Congenital Rubella Syndrome (CRS), with essential criteria and basic principles to verify the elimination process [3]. This plan was implemented by all countries and territories in the Region between 2011 and 2016. Regional verification of the elimination of rubella and CRS occurred in April 2015, with verification of measles elimination occurring in September 2016 [4].

Following the verification of the elimination of both endemic diseases, the Ministers of Health of the Member States approved the implementation of the Action Plan for the Sustainability of Measles, Rubella and CRS Elimination 2018–2023 to set new strategic guidelines to maintain these two public health milestones [5]. Due to the resurgence of measles outbreaks globally which spread to the region between 2018 and 2019, the PAHO Technical Advisory Group on Vaccine-Preventable Diseases (TAG) recommended the creation of a new Regional Commission for the Monitoring and Re-verification of Measles, Rubella and CRS Elimination (RVC) in 2019 [6]. Two countries, Venezuela and Brazil, unfortunately experienced re-established endemic transmission in 2018 and 2019, respectively, from cases imported from other regions. During the Third RVC Annual Meeting held in November 2023, Venezuela was reverified as being free of measles and Brazil was classified as pending reverification. The United States came close to re-establishing endemic circulation, which fortunately was interrupted before 12 months [7].

However, four countries could not be verified as free of measles or rubella and must submit a new report to the RVC in 2024.

As with other parts of the world, in the Americas the COVID-19 pandemic had a substantial negative impact on vaccination coverage and the quality of epidemiological surveillance indicators between 2020 and 2021. A recent analysis of measles mortality in the six WHO regions between 2000 and 2022 estimated that for the Americas, 6.07 million deaths were averted, being the region with the lowest number of cases and deaths reported in these two decades [8].

An important factor highlighted in this report is the strengthening of the national response capacity of many countries and the contribution of the laboratory members of the regional measles–rubella laboratory network.

The regional measles laboratory network began in 1995 with a region-wide training plan implemented for measles IgM capture by Enzyme Immunoassay (EIA), targeting the biggest countries of the region and following the example of the Polio Laboratory Network. This activity was coordinated by the PAHO and facilitated by the US-CDC in Atlanta. In 1996, rubella IgM testing was implemented among specimens with negative IgM measles results. Since 1999 and following the TAG’s recommendation to accelerate rubella and CRS prevention [9], rubella was integrated into the measles surveillance system. Sequential testing for both measles IgM and rubella IgM was implemented in the regional laboratory network [10]. Sequential testing for rubella occurs when measles serology is negative. In addition, viral isolation and molecular testing was realized in many laboratories.

Since 2003, IgM parallel testing for measles and rubella suspected cases was conducted in each laboratory member of the regional network. The implementation of this testing for both measles and rubella simultaneously was facilitated by IgM tests being readily available. The availability of commercial assays of the same manufacturer (Dade Behring Enzygnost) and same EIA type (indirect IgM assay) allowed for rapid response.

By 2011, the PAHO measles and rubella laboratory network had 138 members. The structure of the network comprised 1 global specialized laboratory, 2 regional reference laboratories, 21 national laboratories, and 114 subnational laboratories in eight countries: Argentina, Bolivia, Brazil, Colombia, Ecuador, Mexico, Paraguay, and Venezuela [11]. By 2023, there were 139 measles and rubella laboratories in the region, comprising 1 global specialized laboratory, 2 regional reference laboratories, 20 national laboratories, and 116 subnational laboratories in six countries: 27 in Argentina, 27 in Brazil, 26 in Canada, 3 in Colombia, 2 in Ecuador, and 31 in Mexico.

After the region achieved the verification of the elimination of rubella and measles, a reduction in the notification rate of measles and rubella suspected cases occurred. As a result, some countries unfortunately even reduced the number of subnational laboratories. However, during the last few years, in order to maintain a timely response to the surveillance system, some countries correctly decided to seize the opportunity to increase the number of subnational laboratories to provide a timely response to the program.

## 2. Materials and Methods

This review describes the main regional strategies and their impact on sustaining the elimination of rubella and CRS from 2019 to 2023 and on the recovery of measles-free status.

The sources of information consulted were the PAHO/WHO-ISIS Integrated Measles and Rubella Epidemiological Surveillance System [12], the Joint Reporting Form (eJRF) of the WHO/Unicef [13], Technical Advisory Group (TAG) reports [7], measles–rubella national reference laboratory reports, and national measles–rubella elimination sustainability reports to the Regional Measles–Rubella Elimination Monitoring and Re-verification Commission [14].

The analysis of vaccination coverage indicators and epidemiological surveillance indicators is presented in the tables and one figure. The qualitative analysis is presented with the description of the activities for strengthening the measles–rubella laboratory network, the best practices that the countries applied to improve vaccination coverage, and surveillance indicators.

## 3. Results

### 3.1. Vaccination Coverage in 2019–2022

Regional coverage with the measles, rubella, and mumps (MMR) vaccine has been declining since before the pandemic. In 2018, the coverages of the first (MMR-1) and second doses (MMR-2) were 91% and 84%, respectively. In 2019, they were 87% and 75%, respectively; in 2020, they were 87% and 65%; in 2021, they were 85% and 68%; and in 2022, they were 85% and 71%, respectively (Table 1).

During 2024, PAHO estimated that in 2019 there were 1.4 million 1-year-old children at the regional level that did not receive a timely first dose of MMR-1. In 2020, 1.8 million children did not receive a timely first dose; in 2021, 1.7 million did not, and in 2022, 1.6 million did not. Almost half of the children who did not receive MMR-1 lived in seven countries that accumulated 48% of the total cohort of children aged 12–23 months in the region [15].

Of the 35 PAHO/WHO Member States, in 2019, 16 of the 35 countries achieved 95% coverage with MMR-1 and 9 of 35 with MMR-2. In 2020, only 8 of 35 countries reached 95% coverage with MMR-1 and 5 of 35 with MMR-2. In 2021, only 6 of 35 countries reached 95% coverage with MMR-1 and 2 of 35 with MMR-2. In 2022, this coverage was reached in 9 of 35 countries with MMR-1 and f5 of 35 with MMR-2 (Table 1).

Given the limitations of the Member States in achieving adequate vaccination coverage with second doses of the MMR vaccine, in accordance with the PAHO TAG and RVC recommendations, the countries have taken the opportunity to implement national follow-up vaccination campaigns (VCs), which included vaccination of the entire target population against measles and rubella regardless of their vaccination history (indiscriminate vaccination). These campaigns were monitored for quality, coverage of the target population, efficiency, homogeneity, and timeliness of response. Between 2019 and 2023, 17 countries in Latin America and the Caribbean (LAC) conducted 17 VCs against measles and rubella, vaccinating 47,274,472 children aged 1 to 12 years with MMR and bivalent measles–rubella (MR) vaccines. Of these 16 VCs, seven achieved a coverage of ≥95% and six between 80 and 94%. During and after the COVID-19 pandemic, between 2021 and 2023, 12 countries conducted follow-up vaccination campaigns (VCs), vaccinating 37,479,561 million children in an age range of 1–12 years; five of them targeted the age group of 1–6 years and half of them applied the PAHO’s high-quality criteria and microplanning process. One important fact is that at least seven of the twelve VCs had an impact on the recovery of MMR-1 and MMR-2 coverage. These VCs were conducted in Argentina, Brazil, Bolivia, Colombia, Dominican Republic, Ecuador, El Salvador, Honduras, Mexico, Nicaragua, Paraguay, and Venezuela (Table 2) [13].

### 3.2. Measles Outbreaks in the Period 2019–2023

Large global measles outbreaks between 2018 and 2019 facilitated the importation of the virus to countries in the region. In the Americas, from 2019 to 2023, eighteen countries reported 29,364 imported, import-related, or endemic measles cases, and in two of them, Brazil and Venezuela, endemic transmission was re-established as previously mentioned.

Brazil reported 30,450 confirmed endemic measles cases for this period. Venezuela had an outbreak that started in 2017 from an importation, and in 2019 reported 548 confirmed endemic measles cases. From 2019 to 2023, the United States reported 1515 confirmed measles cases, all imported or related to importation. These three countries accounted for 97.4% of the confirmed measles cases reported in the entire region. Other smaller, well-contained measles outbreaks occurred during this period without re-establishment of endemic transmission: Colombia reported 252 cases in 2019, Mexico reported 216 cases between 2019 and 2020, and Argentina reported a total of 201 confirmed measles cases in the period 2019 to 2022, none with chains of transmission of more than one year. In 2023, only 72 cases were reported in four countries: Canada (12), Chile (1), Costa Rica (1), and the United States (58). Indeed, the year 2023 had the lowest number of confirmed measles cases reported in the history of the region [16,17,18,19] (Figure 1).

In December 2021, the PAHO/WHO conducted a Regional Risk Assessment on Vaccine-Preventable Diseases (VPDs) like Diphtheria, Measles, Yellow Fever, and Polio in the Context of COVID-19. This assessment alerted countries to the very-high-risk potential for other VPD outbreaks due to extremely the low vaccination coverage rates of all antigens, coupled with the weak surveillance performance indicators. This assessment also alerted countries to an increase in migration and population movement across borders during 2022 to 2023, and its potential negative impact on the healthcare systems across the region [16].

### 3.3. Epidemiological Surveillance Indicators between 2019 and 2023

The PAHO/WHO regional measles–rubella surveillance system monitors weekly case-based active rash and fever surveillance in the 21 countries that report directly to the PAHO/WHO through ISIS. The PAHO uses these data to issue a biweekly bulletin to alert all countries to their surveillance compliance. Using data from 2019 to 2023, the PAHO conducted a intensive analysis of compliance with the four strategic lines related to vaccination coverage, epidemiological surveillance indicators, and preparedness for rapid response to outbreaks of these diseases. With gaps defined in surveillance performance defined, the PAHO targets support to countries whose surveillance performance is declining [20].

The notification rate of suspected cases should be of at least 2 cases per 100,000 habitants. In 2019, the rate was 14.13; in 2020 it was 3.93: in 2021 it was 1.44; in 2022 it was 2.32, and in 2023 it was 2.32. Fortunately, this indicator was achieved in all years except for 2021. Table 3 demonstrates a significant reduction in the countries that met each indicator in the study period during the pandemic years (Table 3).

### 3.4. Best Practices for Strengthening the Laboratory Network

Twenty-three measles and rubella laboratories (global specialized, regional reference, and national) participated annually in the Global Measles and Rubella Laboratory Network (GMRLN’s) External Quality Assurance (EQA) exercises for IgM, real-time Reverse-Transcription–Polymerase Chain Reaction (rRT-PCR), and genotyping for both viruses. All the laboratories have IgM and IgG testing capacities, and in addition 13 have rRT-PCR and 9 have IgM, IgG, rRT-PCR, and sequencing capacities.

In the countries that have a subnational laboratory network, the national laboratory is responsible for providing training, quality control, and confirmatory testing, and in some countries it is also responsible for performing the genetic characterization of the virus identified in confirmed measles or rubella cases.

From 2019 to 2023, the PAHO conducted three regional meetings of the measles and rubella laboratory network to review the progress, achievements, and challenges of the network. The intent of these meetings was also to learn about the experiences of countries of the region related to laboratory surveillance, laboratory confirmation, and case classification, and to discuss the new assays available in the GMRLN and quality control.

Because the COVID-19 pandemic diverted all public health attention to the pandemic response, countries found it nearly impossible to advance any of the immunization activities planned. In 2021, the lab network re-started some virtual activities, and then transitioned back to face-to-face training and meetings.

During this period, the PAHO seized the opportunity to deliver a total of five on-site visits, three on-site training courses on rRT-PCR, one virtual training course on rRT-PCR, one hands-on genotyping training course, and one sequence analysis training course to laboratories of the regional network.

In elimination settings, when incidence declines the predictive positive value of IgM testing should be lower, and false positive results will increase. To face this challenge, the PAHO used this opportunity to help enhance laboratory performance. Critically, important additional sampling (second serum specimens, nasal swab, throat swab, urine) and additional testing (IgG seroconversion, IgG-avidity, rRT-PCR, sequencing) is required to improve the certainty of confirming or discarding suspected cases [21]. Countries have embraced this support. Laboratories of the regional network have used this support to improve their technical capacities and capabilities to perform these assays. This collective action among those responsible for epidemiological and laboratory surveillance advances a deeper analysis of special cases, mitigating the risk of reporting false positives.

In addition, the Regional Laboratory Network has contributed to expanding the genetic information of viruses related to confirmed measles cases. During 2019 to 2023, there was a steady uptrend in the proportion of measles sequences reported to the WHO’s database for Measles Virus Nucleotide Surveillance (MeaNS), with a range between 4 and 64% and an average of 6%. Nevertheless, the number of confirmed measles cases substantially decreased, with 23,284 cases reported in 2019 and only 72 cases reported in 2023. Therefore, the uptrend observed in the proportion of measles genotypes could be explained by this significant reduction in confirmed measles cases. In addition, when countries are experiencing large-size outbreaks, such as in Brazil in 2019, the regional recommendation is to test a proportion of confirmed cases for virologic surveillance every two months to monitor if the same genotype and strain are still circulating, or a new importation or introduction has occurred [21]. The latter should be supported by sound epidemiological information.

### 3.5. Best Practices to Increase Vaccination Coverage

Due to the emergence of other public health priorities and an increasingly complex political, social, and epidemiological scenario, as with surveillance, countries are committed to revitalizing their national immunization programs in the post-pandemic period [22].

Colombia is a special example of a country that has maintained good vaccination coverage, outstanding performance related to surveillance indicators, and timely responses to outbreaks. Colombia reported the following progress to the RVC during the Third Annual Meeting in 2023: In the VC carried out in Colombia between 2021 and 2022 aimed at children between 1 and 10 years of age, the country reached a target of 95%, vaccinating 7,208,981 children. During the same period, 680,463 one-year-old children were vaccinated with an additional dose of MR and 335,920 children aged 2 to 11 years were vaccinated with a zero dose of MMR-1. The Ministry of Health continued the search for the susceptible population after the VC by conducting mop-up and active searches for the missing children to be vaccinated from 3 to 13 years of age, vaccinating 63,240 children.

The country also implemented a vaccination plan for migrants coming into the country and for Colombian travelers bound for Europe and other regions of the world. These efforts were intended to help increase population immunity and to prevent the spread of the virus in the event of importations.

## 4. Discussion

Between 2020 and 2023, the WHO declared a public health emergency of international concern (PHEIC) for COVID-19. The pandemic imposed immense challenges to global health. Undoubtedly, the COVID-19 pandemic had a negative impact on MRE strategies worldwide and specifically in the Americas, with a decline in MMR vaccination coverage and surveillance indicators. The decline in coverage was most notable with the first dose (MMR-1), which stagnated at 85% in 2022–2023. In contrast, there was a slow and sustained recovery in coverage with the second dose (MMR-2), with 68% and 70% rates in 2021 and 2022.

An opportunity to improve MMR coverage is conducting VCs when linked to essential immunization services. Together, these strategies with essential immunization taking the lead can potentially increase population immunity everywhere, which is needed to ultimately stop measles virus transmission. Those countries that applied the PAHO’s high-quality criteria and microplanning in their VCs achieved the goal ≥ 95%. This confirms once again that VCs are a key intervention for sustaining measles and rubella elimination in our region when they have been organized with high-quality criteria and a good microplanning process.

The COVID-19 pandemic negatively impacted the notification of suspected cases, reducing the sensitivity of surveillance. The work force was directed to focus on the pandemic, so measles–rubella surveillance suffered. In 2020 and 2021, all indicators dropped, but the important thing is that the countries began to recover their suspect case notification rates from 2022 and 2023.

The performance of the surveillance indicators was relatively better in the Central American and Latin Caribbean countries, followed by the South American countries and finally the English-speaking Caribbean countries. From 2023 onwards, an increase in the number of countries complying with the suspect case notification rate occurred in all subregions.

In countries that had active outbreaks in 2020, a decrease in confirmed measles cases occurred, possibly due to the social and physical distancing measures that prevented virus circulation.

In terms of the lessons learned from the countries, Colombia implemented a high-quality follow-up vaccination campaign during 2021 and 2022 targeting children aged 1–11 years based on the available epidemiologic data. Colombia acted on the opportunity to strengthen and contain the decline in the coverage of the other biologics in the children’s immunization schedule. Their approach raised awareness among parents to ensure their children’s adherence to the national immunization schedule.

Increasingly important is the need to be able to document the viral genotypes and lineages associated with infection [23]. Molecular epidemiological data are of great value in documenting importations, especially when there are multiple importations of measles in a defined area, especially over a short period of time.

## 5. Conclusions

The countries of the region must concentrate their efforts on maintaining achievements in immunization, irrespective of the targeted disease. Failure to recover good vaccination coverage could have negative repercussions on the sustainability of regional MRE, given the active outbreaks in the countries of the other five regions of the world. Nevertheless, the countries of the Americas are clearly committed to sustaining MRE despite the challenges they face, as evidenced by the fact that currently no country has endemic virus circulation, and the region is once again moving towards regaining the status of a region free of endemic measles virus.

The countries must remain vigilant, improve the sensitivity of the surveillance system and be prepared for a rapid response due to the likelihood of importation of measles and rubella cases. For any suspect case, a serum specimen, nasopharyngeal swabs, and/or urine specimens should be collected for laboratory testing and ultimately for timely case confirmation.

## Figures and Tables

**Figure 1 vaccines-12-00690-f001:**
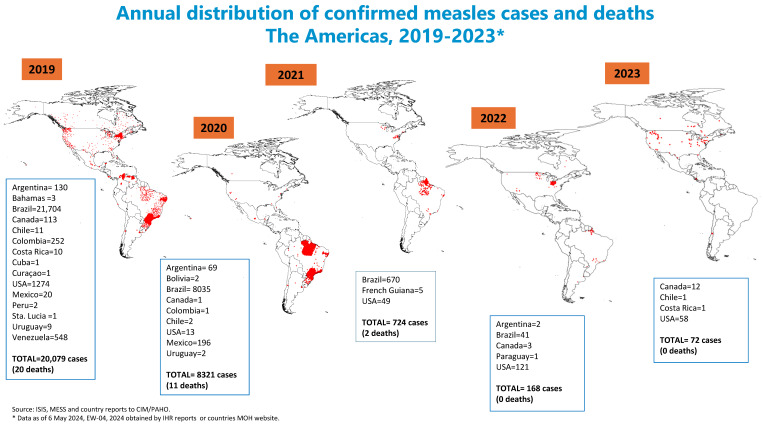
Annual distribution of confirmed measles cases and deaths The Americas, 2019–2023 *.

**Table 1 vaccines-12-00690-t001:** MMR vaccination schedule, estimated coverage with the first and second doses of measles-containing vaccines *, number of confirmed measles cases **, and confirmed measles incidence § by country or area—Pan American Health Organization/World Health Organization.

	MMR Schedule ¶		2019			2020			2021			2022		
	First Dose Age, mos	Second Dose Age, mos	Coverage, %		No. of Measles Cases ** (Incidence) §	Coverage, %		No. of Measles Cases ** (Incidence) §	Coverage, %		No. of Measles Cases ** (Incidence) §	Coverage, %		No. of Measles Cases ** (Incidence) §
Member States			MMR1	MMR2		MMR1	MMR2		MMR1	MMR2		MMR1	MMR2	
**The Americas**			**87**	**75**	**24,079 ** (17.7)**	**87**	**65**	**8321 (6.07)**	**85**	**68**	**724 ** (0.53)**	**85**	**71**	**168 (0.12)**
**North America**			**90**	**94**	**1387 (3.78)**	**91**	**83**	**14 (0.04)**	**92**	**83**	**49 (0.13)**	**92**	**79**	**124 (0.33)**
Canada	12		90	87	113 (3.02)	90	83	1 (0.03)	90	83	0 (0)	92	79	3 (0.08)
United States	12	48	90	95	1274 (3.87)	91		13 (0.04)	92		49 (0.15)	92		121 (0.36)
**Mexico**	12	18–72	**73**	**73**	**20 (0.16)**	**100 ††**	**83**	**196 (1.52)**	**100 ††**	**97**	**0 (0)**	**86**	**83**	**0 (0)**
**Central America**			**91**	**86**	**10 (0.2)**	**86**	**79**	**0 (0)**	**86**	**75**	**0 (0)**	**83**	**73**	**0 (0)**
Costa Rica	15	48	95	100	10 (1.98)	95	81	0 (0)	89	69	0 (0)	90	75	0 (0)
Guatemala	12	18	90	78	0 (0)	88	79	0 (0)	81	72	0 (0)	83	69	0 (0)
El Salvador	12	15–18	82	87	0 (0)	71	56	0 (0)	85	70	0 (0)	65	59	0 (0)
Honduras	12	18	89	85	0 (0)	82	79	0 (0)	81	75	0 (0)	77	71	0 (0)
Nicaragua	12	18	100 ††	100	0 (0)	100 ††	100 ††	0 (0)	100	86	0 (0)	100 ††	100 ††	0 (0)
Panama	12 ¶	18 ¶	97	97	0 (0)	80	74	0 (0)	92	75	0 (0)	87	75	0 (0)
**Latin Caribbean**			**85**	**59**	**1 (0.03)**	**86**	**59**	**0 (0)**	**83**	**60**	**0 (0.13)**	**89**	**71**	**0 (0)**
Cuba	12	72	100	100	1 (0.09)	98	99	0 (0)	100	100	0 (0)	100	100	0 (0)
Dominican Republic	12	18	96	60	0 (0)	82	55	0 (0)	88	60	0 (0)	100 ††	87	0 (0)
Haiti	9 *	12–23 *	70	41	0 (0)	83	46	0 (0)	73	44	0 (0)	76	46	0 (0)
**Andean**			**89**	**62**	**802 (5.72)**	**82**	**61**	**3 (0.02)**	**77**	**63**	**0 (0)**	**79**	**66**	**0 (0)**
Bolivia (Plurinational State of)	12	18–23	79	44	0 (0)	74	46	2 (0.17)	75	56	0 (0)	69	49	0 (0)
Colombia	12	60	95	89	252 (5.01)	91	88	1 (0.02)	86	86	0 (0)	88	84	0 (0)
Ecuador	12	18	83	76	0 (0)	81	70	0 (0)	65	58	0 (0)	74	60	0 (0)
Peru	12	18	85	66	2 (0.06)	77	52	0 (0)	78	60	0 (0)	74	54	0 (0)
Venezuela (Bolivarian Republic of)	12 ¶	18 ¶	93	13	548 (19.22)	76	28	0 (0)	68	38	0 (0)			0 (0)
**Brazil**	12		**91**	**54**	**21,704 (102.84)**	**80**	**44**	**8035 (37.8)**	**73**	**46**	**670 (3.13)**	**81**	**58**	**41 (0.19)**
**Southern Cone**			**87**	**84**	**150 (2.02)**	**79**	**73**	**73 (0.97)**	**84**	**79**	**0 (0)**	**82**	**84**	**3 (0.04)**
Argentina	12	60	86	84	130 (2.9)	77	71	69 (1.53)	86	79	0 (0)	85	94	2 (0.04)
Chile	12	36	95	91	11 (0.58)	91	83	2 (0.1)	92	91	0 (0)	94	81	0 (0)
Paraguay	12	48	75	71	0 (0)	68	60	0 (0)	56	55	0 (0)	42	41	1 (0.15)
Uruguay	12	15	96	99	9 (2.6)	96	92	2 (0.58)	93	82	0 (0)	96	92	0 (0)
**non-Latin Caribbean**			**94**	**87**	**4 (0.63)**	**90**	**84**	**0 (0)**	**87**	**80**	**0 (0)**	**91**	**84**	**0 (0)**
Antigua and Barbuda	12	18	97	71	0 (0)	89	80	0 (0)	85	76	0 (0)	100	100	0 (0)
Bahamas	12	15	83	83	3 (7.7)	87	83	0 (0)	82	80	0 (0)	80	65	0 (0)
Belize	12	18	96	95	0 (0)	82	87	0 (0)	79	77	0 (0)	81	77	0 (0)
Barbados			100 ††	85	0 (0)	89	78	0 (0)	77	70	0 (0)	85	73	0 (0)
Dominica	12	18	92	92	0 (0)	92	90	0 (0)	88	87	0 (0)	87	87	0 (0)
Grenada	12	18	94	82	0 (0)	83	79	0 (0)	70	55	0 (0)	76	68	0 (0)
Guyana	12	18	98	92	0 (0)	100 ††	97	0 (0)	95	83	0 (0)	100 ††	100 ††	0 (0)
Jamaica	12	18	94	92	0 (0)	93	89	0 (0)	88	85	0 (0)	91	83	0 (0)
Saint Kitts and Nevis	12	18	97	98	0 (0)	95	99	0 (0)	96	94	0 (0)	95	93	0 (0)
Saint Lucia	12	18	96	75	1 (5.47)	89	72	0 (0)	77	66	0 (0)	81	63	0 (0)
Saint Vincent and the Grenadines	12	18	100 ††	100 ††	0 (0)	100 ††	100 ††	0 (0)	96	94	0 (0)	99	100	0 (0)
Suriname	12	18	85	43	0 (0)	66	35	0 (0)	79	58	0 (0)	95	65	0 (0)
Trinidad and Tobago	12	24	99	92	0 (0)	91	90	0 (0)	93	88	0 (0)	92	92	0 (0)

Abbreviations: MMR = measles-mumps and rubella vaccine; MMR1 = first MMR dose; MMR2 = second MMR dose; NA = not applicable; NR = not reported; WHO = World Health Organization. * According to OFFICIAL coverage reported. For MMR1, among children aged 1 year or, if MMR is given at age ≥1 year, among children aged 24 months. For MMR2, among children at the recommended age for administration of MMR2, per the national immunization schedule. Haiti administers MR vaccine. The data were last revised on 15 July 2023 at https://immunizationdata.who.int. ** Includes cases confirmed by laboratory testing or epidemiologic linkage and clinically compatible cases. Clinically compatible cases met the WHO measles clinical case definition, had no adequate specimen collected, and could not be epidemiologically linked to a laboratory-confirmed case of measles. Case totals based on country reports to CIM-PAHO weekly surveillance reports. WHO Non-member states Curaçao and French Guiana reported measles cases, 1 (2019) and 5 (2022) respectively. § Cases per 1 million population. ¶ MMR schedule is the 2022 schedule; Panama and Venezuela schedule from 2021. †† coverage data >100%. The data of each subregion have been bolded to highlight the average of all countries belonging to that subregion.

**Table 2 vaccines-12-00690-t002:** Follow-up campaigns and impact on the administrative vaccination coverage with MMR-1 and MMR-2. Americas 2021–2023.

**Countries with Follow-up Campaigns 2021**	**Age group**	**Vaccine**	**Target group**	**Vaccinated children**	**Coverage VC** **Follow-up**	**MMR-1 (%) 2020**	**MMR-2 (%) 2020**	**MMR-1 (%) 2021**	**MMR-2 (%) 2021**	**MMR-1 (%) change**	**MMR-2 (%) change**	**Use high-quality criteria (Y/N)**	**Applied micro planning (Y/N)**
**Mexico**	1–9 yo	MR/MMR	11,662,394	11,202,785	**96**	100	83	100	97	0	14	Yes	Yes
**Colombia**	1–10 yo	MR/MMR	7,552,514	7,208,981	**95**	91	88	86	86	−5	−2	Yes	Yes
**Bolivia**	1–6 yo	MR/MMR	1,181,729	1,039,922	88	74	46	75	56	1	10	No	No
**Paraguay**	1–5 yo	MMR	738,619	535,703	73	68	60	56	51	−12	−9	No	No
**Total children vaccinated in 2022**			**21,135,256**	**19,987,391**	95								
**Countries with Follow up Campaigns in 2022**	**Age group**	**Vaccine**	**Target group**	**Vaccinated children**	**Coverage VC** **Follow-up**	**MMR-1 (%) 2021**	**MMR-2 (%) 2021**	**MMR-1 (%) 2022**	**MMR-2 (%) 2022**	**MMR-1 (%) change**	**MMR-2 (%) change**	**Use high-quality criteria (Y/N)**	**Applied micro planning (Y/N)**
**Argentina**	13–59 m	MMR	2,315,692	1604631	69	86	79	85	94	−1	15	No	No
**Brazil**	1–4 yo	MR/MMR	12,927,057	6,552,277	51	73	46	81	58	8	12	No	No
**Dominican Republic**	1–5 yo	MR/MMR	951,554	934,329	**98**	88	60	100	87	12	27	Yes	Yes
**El Salvador**	1–6 yo	MMR	644,089	592,562	92	85	70	65	59	−20	−11	No	No
**Honduras**	1–6 yo	MMR	1,669,999	1,386,099	83	81	75	77	71	−4	−6	No	No
**Nicaragua**	1–6 yo	MMR	723,790	823,127	**114**	100	96	100	100	4	0	Yes	Yes
**Venezuela**	1–6 yo	MR/MMR	2,811,860	2,479,675	88	68	38	N/D	N/D	N/D	N/D	No	No
**Total children vaccinated in 2022**			**22,044,041**	**14,372,700**	65								
**Countries with Follow up in 2023**	**Age group**	**Vaccine**	**Target group**	**Goal**	**Coverage**	**MMR-1 2022**	**MMR-2 2022**	**MMR-1 2023 ***	**MMR-2 2023 ***	**MMR-1 (%) change**	**MMR-2 (%) change**	**Use high-quality criteria (Y/N)**	**Applied micro planning (Y/N)**
**Ecuador ***	1–12 yo	MR/MMR	3,189,901	3,119,470	**98**	74	60	97	81	23	21	Yes	Yes
**Total children vaccinated in 2023**			**3,189,901**	**3,119,470**	98								
**Total children targeted and vaccinated**			**46,369,198**	**37,479,561**	**81**								

* Preliminary data eJRF. Abbreviations: MR = measles-rubella vaccine; MMR = measles-mumps-rubella vaccine; MMR1 = first MMR dose; MMR2 = second MMR dose; VC = vaccination campaign; yo = years old; m = months. Source: Country reports through the electronic PAHO-WHO/UNICEF Joint Reporting Form (eJRF), 2020–2023 and country reports to the Measles Rubella Elimination Regional Commission. Bold numbers are those percentages ≥ 95% of vaccination coverage.

**Table 3 vaccines-12-00690-t003:** Measles and rubella surveillance indicators performance *.

	Number and Percentage of Countries Meeting the Indicators				
MR ** Surveillance Indicators Quality	2019	2020	2021	2022	2023
Annual suspected MR ** case rate per 100,000 population	16 (76)	7 (33)	6 (29)	10 (48)	10 (48)
Percentage of MR ** suspected cases with adequate case investigation	14 (67)	13 (62)	12 (57)	14 (67)	13 (62)
Percentage of suspected cases with adequate serum specimen collected, tested in a proficient laboratory	19 (90)	16 (76)	16 (76)	19 (90)	14 (67)
Percentage of suspected cases with serum specimens sent to laboratory within 5 days	11 (52)	10 (48)	10 (48)	11 (52)	13 (62)
Percentage of IgM laboratory results reported to national public health authorities within 4 days	14 (67)	11 (52)	12 (57)	15 (71)	13 (62)
Regional rate of MR suspected cases by 100,000 habitants	14.13	3.93	1.44	2.32	2.32

* Data from 21 countries and 13 countries of English Caribbean Subregion reporting to PAHO (except Canada and USA). ** MR = measles rubella.

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
