# Peer review of "Sustaining the Elimination of Measles, Rubella and Congenital Rubella Syndrome in the Americas, 2019–2023: From Challenges to Opportunities"

_vaccines, 2024, doi:10.3390/vaccines12060690_

Round 1

Reviewer 1 Report

Comments and Suggestions for Authors

Dear Authors

thank you for this important contribution to knowledge on recent improvement on the control of measles and rubella in the Americas, it is of high interest and well done

Author Response

Dear reviewer,

Thank you for giving us your feedback on the article. It is very pleasant to have your opinion as an expert on the subject on the contribution that this article will give to the memory of achievements in epidemiology and global public health, such as the elimination of measles, rubella and congenital rubella syndrome in the region of the Americas.

Reviewer 2 Report

Comments and Suggestions for Authors

The review manuscript by Rey-Benito et al. summarizes the recent country status of measles, rubella and CRS elimination in the WHO American Region, based on data such as vaccination coverage and number of reported cases. This manuscript provides a good overview of the pioneering efforts to eliminate measles, rubella and CRS in the Americas and the challenges to maintaining this status, thus providing useful information to promote a global control plan for these infectious diseases.

However, very minor modifications may be required in the following points.

1)     Line 31: In the Abstract, it is stated that "One of the countries that had lost certification of elimination in 2018 managed to be reverified in 2023; the other is pending reverification.” However, I cannot find such a statement in the text. Please include that description in the text or delete it from the abstract.

2)     3.1. Vaccination coverage in 2019-2022
There may be some discrepancies between the following statements and those in Table 1. Please check and correct if necessary.
Line 130; “in 2022 they were 85% and 70% respectively.” But 71% in Table 1.
Line 136; “Of the 35 PAHO/WHO Member States, in 2019 14 of the 35 countries achieved 95% coverage with MMR-1 and eight of the 35 with MMR-2.” But in Table 1 they appear to be 16 and 9 countries respectively.

3)     “Figure" or "Figure 1" should be placed at the beginning of the legend in the figure. There is a term “Figure 1" in Line 182.

4)     There may be some discrepancies between the description in the text and in the Figure. Please check and correct if necessary.
Line 170; “In the Americas, from 2019 to 2023, 18 countries reported 32 569 imported, import-related or endemic measles cases and in two of them, Brazil and Venezuela, endemic transmission was re-established as previously menioned.” The figure shows a total of 29,364 cases.
Line 216; “Mexico 216 cases in 2020”. The figure shows Mexico = 196.

5)     Line 220; “The notification rate of suspected cases at the regional level was met in all years of the study period, except in 2021, which dropped to 1.4 x 100,000 inhabitants, but fortunately this indicator recovered in 2022 and 2023, when the Region met the expected rate.”
Please explain the expected rate. It is >2.0 x 100,000 inhabitants, right? Also, it would be easier to understand the degree of achievement against the target if the rates for other years were described, not just 2021. How about incorporating them into Table 3?

6)     Line 223, “Table 2 demonstrates a significant reduction in the countries that met each indicator in the study period during the pandemic years (Table 3).” Please correct the underlined parts.

7)     Several abbreviations, EIA, SRC, RVC, rRT-PCR, MRE, have appeared without explanation. Please explain them appropriately.

Author Response

Thank you for giving us your feedback on the article. It is very pleasant to have your opinion as an expert on the subject on the contribution that this article will give to the memory of achievements in epidemiology and global public health, such as the elimination of measles, rubella and congenital rubella syndrome in the region of the Americas.
Below you can find the response to your suggestions to improve the article and we appreciate having detected the inconsistencies to correct them. Likewise, we have taken advantage of this review process to update table 2 on Vaccination Campaigns with more updated data extracted from the WHO/UNICEF Joint Reporting Form (JRF).

1)     Line 31: In the Abstract, it is stated that "One of the countries that had lost certification of elimination in 2018 managed to be reverified in 2023; the other is pending reverification.” However, I cannot find such a statement in the text. Please include that description in the text or delete it from the abstract.

R/ We have added a clarification about Venezuela and Brazil status of classification in the introduction section.

2)     3.1. Vaccination coverage in 2019-2022
R/ We changed MMR2 coverage from 70 to 71% as shown in the table 1 for 2022 and we corrected numbers in the line 138 and 139.

3) Figure" or "Figure 1" should be placed at the beginning of the legend in the figure. There is a term “Figure 1" in Line 182.

R/ It is done

4) There may be some discrepancies between the description in the text and in the Figure. Please check and correct if necessary.
Line 170; “In the Americas, from 2019 to 2023, 18 countries reported 32 569 imported, import-related or endemic measles cases and in two of them, Brazil and Venezuela, endemic transmission was re-established as previously mentioned.” The figure shows a total of 29,364 cases. Line 216; “Mexico 216 cases in 2020”. The figure shows Mexico = 196.

R/ Thanks for this correction. This mistake was done because at the beginning number of cases corresponded to the period 2018 - 2023 but at the end we decided to cover the period between 2019-2023. This change was not saved appropriately.

5) Line 220; “The notification rate of suspected cases at the regional level was met in all years of the study period, except in 2021, which dropped to 1.4 x 100,000 inhabitants, but fortunately this indicator recovered in 2022 and 2023, when the Region met the expected rate.”
Please explain the expected rate. It is >2.0 x 100,000 inhabitants, right? Also, it would be easier to understand the degree of achievement against the target if the rates for other years were described, not just 2021. How about incorporating them into Table 3?

R/ The suggestion to clarify the parameter of the notification rate of at least 2 x 100,000 habitants has been included and also the regional notification rates by each year. We have also incorporated these regional rates at the bottom of a new table 3.

6) Line 223, “Table 2 demonstrates a significant reduction in the countries that met each indicator in the study period during the pandemic years (Table 3).” Please correct the underlined parts. 

R/ Number of table 2 was corrected, and now is table 3.

7) Several abbreviations, EIA, SRC, RVC, rRT-PCR, MRE, have appeared without explanation. Please explain them appropriately.

R/ It is done.
